# Single-Port Laparoscopic Hepatectomy: Slovenian Single-Center Experience

**DOI:** 10.3390/diseases13060187

**Published:** 2025-06-18

**Authors:** Jerica Novak, Miha Petrič, Blaž Trotovšek, Mihajlo Đokić

**Affiliations:** 1Department of Abdominal Surgery, Ljubljana University Medical Center, Zaloška cesta 7, 1000 Ljubljana, Slovenia; miha.petric@kclj.si (M.P.); blaz.trotovsek@kclj.si (B.T.); mihajlo.djokic@kclj.si (M.Đ.); 2The Faculty of Medicine, University of Ljubljana, Vrazov trg 2, 1000 Ljubljana, Slovenia

**Keywords:** hepatectomy, minimally invasive hepatobiliary surgery, single-port laparoscopy

## Abstract

Background: Single-port laparoscopic hepatectomy is a minimally invasive modality for the treatment of benign and malignant liver lesions. Due to the method’s technical challenges, it is suitable for experienced hepatobiliary surgeons and selected groups of patients. The aim of this study was to evaluate the results of a single Slovenian center performing single-port laparoscopic hepatectomy with a literature overview. Methods: A single-center retrospective consecutive case series of the twenty-six patients with liver disease operated with the single-port technique from January 2018 to July 2024 at the Department of Abdominal Surgery at the University Medical Centre, Ljubljana, was performed. Lesions were located in easy-to-treat segments. Operative time, conversion rate, length of hospital stay, and surgical complications were recorded and evaluated. Results: We performed twenty-six single-port laparoscopic liver resections (median age 63.5, range 31 to 79 years). The mean operative time was 92 ± 31 min. None of the cases were converted to multi-port laparoscopic or open surgery. Safe resection margins were obtained in cases of malignant disease. The mean hospital stay was 4 days. The post-operative complication rate involving intervention was 7% (2/26). The incisional hernia rate was 11.5% (3/26). No life-threatening surgical complications or morbidity were noted. Conclusions: Single-port laparoscopic hepatectomy is a safe and feasible technique for the resection of benign and malignant liver lesions in the hands of skilled and well-trained hepatobiliary surgeons.

## 1. Background

Laparoscopic surgical procedures have become routine procedures in almost all fields of abdominal surgery and have, in many cases, replaced traditional open surgical procedures [1,2]. Due to the complexity of the liver anatomy and technical difficulties during open hepatobiliary surgery, minimally invasive surgery approaches have been adopted slowly and with caution [3,4]. With the progress of technology, as well as surgical technique and skills, laparoscopic and robotic liver surgery have, in recent years, become safe and effective modalities for the treatment of both benign and malignant liver disease [5]. In high-volume hepatobiliary centers with experienced surgeons, not only in the hepatobiliary field but also in the field of advanced minimally invasive surgery, laparoscopic liver resections can account for up to 60% of all liver resections [1]. It is believed that minimally invasive liver resection, compared to open procedures, can significantly reduce the duration of hospital stay, and moreover is related to lower complication rates [1].

With the further progression of minimally invasive surgery, the aim is to minimize the number and size of abdominal incisions; therefore, single-port laparoscopic hepatectomy (SPLH) has been arising as a new surgical method for the treatment of liver disease, with promising results from published reports [6,7,8,9,10,11,12,13,14,15,16,17,18,19,20,21,22,23]. SPLH is technically challenging, as adequate technical equipment and surgeon’s expertise in advanced minimally invasive surgery are necessary to achieve comparable results to multiport laparoscopic liver resection [19]. Due to the narrow surgical view, the management of potential complications, such as intraoperative bleeding and bile leakage, can be demanding [19].

The aim of the present study was to evaluate the results of a single-center SPLH experience in Slovenia, with the priority being to assess the safety, efficiency, and benefit of the procedure for patients with malignant and benign liver disease. An overview of the literature was performed to evaluate our results in comparison to other published studies.

## 2. Methods

### 2.1. Patients

A retrospective chart review of the patients with liver lesions treated at the Department of Abdominal Surgery at the University Medical Centre, Ljubljana, was performed. The object of the study was twenty-six patients (median age 63.5, range 31 to 79 years) with liver lesions who were operated at the Department of Abdominal Surgery between January 2018 and July 2024, and in whom SPLH was performed. Operative time, transfusion rate, conversion rate, ASA physical status score, BMI, length of hospitalization, and surgical complications according to the Dindo–Clavien classification were recorded. Moreover, during follow up, the incisional hernia rate and possible disease recurrence and/or progression were also reported. The clinical features of the study population and surgical procedures performed are described in Table 1.

Approval for the study was obtained from The Medical Ethics Committee of the Republic of Slovenia (Protocol Review Board (MZ 0120-557/2020/11)). Written informed consent was obtained from all of the included patients.

### 2.2. Operative Technique

SPLH was performed by one experienced hepatobiliary and liver transplantation specialist with additional expertise in advanced minimally invasive surgery. The single-port technique was used to treat both benign and malignant disease (detailed information shown in Table 1). Liver lesions were chiefly located in easy-to-treat liver segments, such as the left lateral section or sixth liver segment (detailed information shown in Table 2). No major hepatectomy was performed.

Patients were operated on under general anesthesia and placed in the supine position. In all patients, the Gel POINT advanced access platform (Applied Medical, Rancho Santa Margarita, USA) was used, as no other platform is currently available at our institution. The platform was applied to the abdomen through a four-centimeter-long midline transumbilical laparotomy (Figure 1). Pneumoperitoneum was established with the insufflation of carbon dioxide. An intra-abdominal pressure of 12 mmHg was maintained during the procedure. Three ports (two of them as working ports, respectively) were inserted through the platform. In selected cases, the placement of the additional working port was required to ensure proper triangulation, management of the adhesions, sufficient retraction, and liver preparation and resection (Figure 2). The additional port was then used for drainage placement (detailed information shown in Table 2, respectively). Laparoscopic intraoperative ultrasound was used in cases with preoperatively confirmed malignant disease and in cases of solid liver lesions to provide adequate resection margins. Adhesiolysis, mobilization of the liver, and the transparenchymal phase of the operation were performed with a LigaSure^TM^ device (Medtronic, Minneapolis, MN, USA) and a cavitron ultrasonic surgical aspirator (CUSA). At our institution, liver transections in both open and laparoscopic liver procedures are usually performed with CUSA and LigaSure^TM^ devices. In our opinion, the visualization of bile ducts and vessels is therefore more precise. With the optimal visualization of the sealing of structures, consequently the risk of major bleeding and bile leakage is minimized. Moreover, a vascular stapler was used for the dissection of larger vessels and bile ducts. A TachoSil fibrin sealant patch (Takeda, Linz, Austria) was placed on the liver resection plane. Specimens were retrieved through the midline incision. All specimens were sent for definitive histopathological examination.

### 2.3. Statistical Analysis

The association between categorical variables was tested by the Pearson chi-square test or Fisher’s exact test, as appropriate. All comparisons were two-sided and the *p*-value < 0.05 was considered statistically significant. Statistical analyses were performed using SPSS software (IBM Corp., version 22.0 Armonk, NY, USA).

## 3. Results

We performed SPLH in twenty-six patients (median age 63.5, range 31 to 79 years). In the study population, there were fifteen female (58%) and eleven male patients (42%). Adhesiolysis was performed when necessary. In six cases, adhesions were due to previous intraabdominal procedures (cholecystectomy, gastric sleeve resection, low anterior resection of the rectum, hysterectomy, and laparoscopic colon resection), and in one case due to a large hydatid cyst. The mean operative time was 92 ± 31 min. None of the cases were converted to multiport laparoscopic or open surgery. All the specimens were extracted through the midline laparotomy. No extension of the laparotomy or additional abdominal incision was needed.

Among twenty-six patients, in seven patients resection was due to malignant disease (Table 1). In all of the malignant disease cases, R0 resection was achieved. In four patients, the resection was due to a primary liver tumor (Table 1). In patients with hepatocellular carcinoma, liver cirrhosis was recorded as Child A cirrhosis using the Child–Pugh scoring system.

According to the ASA physical status classification system, 81% of patients were perioperatively classified as ASA 2 and 19% as ASA 3, respectively. Intraoperatively, no transfusion of red blood cells, fresh frozen plasma, or thrombocytes was administrated. The estimated median intraoperative blood loss was 20 ± 56 mL.

In this cohort, no life-threatening intra-operative or postoperative complications during the 90-day follow-up period were recorded. Additionally, in the observed cohort, no mortality was noted. When analyzing post-operative complications, a surgical complication of grade IIIa on the Clavien–Dindo Scale was reported in 7% (2/26) of patients. In both patients, the complication was related to a biliary leak. One biliary leak was due to a leak from a fall-off clip on a cystic duct in a simultaneous liver resection and cholecystectomy. In the second case, the leak was due to a biliary leak after the deroofing of a giant simple biliary cyst (Table 1). In both cases, the leak was successfully resolved with ERCP and choledochal stenting, respectively.

After surgery, all included patients were transferred to the surgical ward and early enteral feeding was started. The length of main hospital stay was 4 ± 1 days. Abdominal drainage systems were placed in all treated patients. The drain was removed on the first post-operative day for all patients except the two patients with biliary leakage.

All patients are still alive and under surveillance. During surveillance, the recorded incisional hernia rate was 11.5% (3/26). In one patient a mesh hernioplasty was performed, but the other two declined hernia repair. A statistically significant correlation between incisional hernia rate and the postoperative complication of biliary leakage was reported (*p* = 0.009). No other factors, demographic or operative (age, sex, BMI, ASA score, previous operations), were significantly associated with the occurrence of incisional hernia in our cohort.

## 4. Discussion

Over the last few decades, especially with the introduction of laparoscopy, surgery has become progressively less invasive [1]. With the tendency to further minimize the invasiveness of surgical procedures, single-port laparoscopic surgery was introduced to the field of abdominal surgery [24]. Although the technique has been safely used for different, less complex abdominal procedures, such as cholecystectomy, appendectomy, inguinal hernia repair, and colectomy, studies on single-port laparoscopic liver resections are still limited [25,26,27,28,29]. The aim of the present study was to evaluate the results of a single Slovenian center performing SPLH, the method that can, in our opinion, be safely used for the treatment of benign and malignant liver lesions in selected patients, mainly when lesions are located in easy-to-treat liver segments.

Some retrospective studies comparing single-port versus multiport laparoscopic hepatectomy (MPLH) have already been performed [19,20,21,23]. The number of included patients undergoing SPLH is generally small, ranging from just a few patients to 155 patients [19,20,21,22,23]. As the procedure is technically difficult, mainly due to the loss of triangulation and narrow surgical view, it is not suited for every patient with liver disease [19,20,21,22,23]. According to these studies, SPLH can be safely used to perform left lateral sectionectomy and partial hepatectomy of easy-to-treat liver segments in cases of benign and also malignant liver disease [18,19,20,21,22,23]. In the largest published study of Han and associates, SPLH was also performed for major hepatectomies with the resection of more than two liver segments [19].

In this cohort, SPLH was performed for treating benign and malignant lesions mainly in the left lateral liver segment and sixth liver segment. Moreover, only the resection of two or less than two liver segments was performed, which is in concordance with other studies [18,19,20,21,22,23]. Compared to other studies, all the operations were performed by one experienced hepatobiliary surgeon. Similar to other authors, we also experienced that SPLH is technically more challenging than MPLH, mainly due to the loss of triangulation, as well as ergonomic difficulties for both the surgeon and the assistant. Therefore, corresponding to the literature, the placement of an additional port was needed in selected cases to provide adequate triangulation for a safe surgical procedure [20]. Moreover, an additional port was also needed to effectively perform adhesiolysis in patients with extended adhesions after previous surgeries, and moreover for safe preparation in cases of large cystic formations. The site was then used for the drainage placement.

Studies comparing SPLH to MPLH reported a significantly shorter operation time in the SPLH group [19,20,21]. Reported mean operative times were comparable to our results (ranging from 113 min to 137 min compared to 92 min, respectively) [19,20,21]. Moreover, comparing SPLH to MPLH, the authors also found significantly smaller blood loss, faster enteral feeding, and shorter length of stay in the SPLH group [19,21,23]. The mean intraoperative blood loss in this cohort was 20 ± 56 mL, with no transfusion of red blood cells, fresh frozen plasma, or platelets required. According to the literature, the blood transfusion rate was smaller in the SPLH group compared to the MPLH group. The reasoning behind this smaller blood loss is probably the fact that in the SPLH group, resection was performed in easy-to-treat segments with less expected technical difficulties [19]. Enteral feeding and locomotor physical therapy in our cohort were started on the first postoperative day. The mean length of hospitalization was comparable to the results of other studies (ranging from 4 to 7 days compared to 4 days, respectively) [19,20,21,22].

Related to studies where lesions from the left lateral segments and sixth and/or fifth liver segments were removed, no conversion to multiport laparoscopic surgery or open surgery was needed [20,22,23]. In comparison, the conversion rate in Han’s group was higher (22.6% for SPLH and 19.8% for MPLH, respectively). Nevertheless, when comparing to Han’s cohort, one must keep in mind that in this group the laparoscopic approach was favored for all hepatic resections, including major ones; therefore, a higher conversion rate is an expected result [19]. The main reasons for conversion in the SPLH group were bleeding and technical failure (60% and 14.3%, respectively) [19]. In this study, no life-threatening intraoperative surgical complications were noted; moreover, the estimated mean blood loss was less than 50 mL. On the other hand, the recorded post-operative complication rate necessitating intervention was 7% (2/26). All major post-operative complications were related to biliary leaks, but these were swiftly recorded, mainly due to the fact that drainage was placed in all treated patients. Both patients were treated due to a lesion in the left lateral segment. In one patient, cholecystectomy was also performed. The sources of leakage were the resection plane and cystic stump, respectively. As the drainage was in place, the copious biliary drainage was noted early in the postoperative course and the complications were successfully treated and did not significantly prolong the hospital stay of the patients. The complication rate in other published studies in SPLH groups are low and similar to MPLH groups [19,21]. Although left lateral sectionectomy is considered less technically challenging, complications do still occur. One consider that the narrow surgical view can impact the management of potential complications such as bile leakage during SPLH; therefore, the importance of meticulous duct ligation is important to prevent major post-operative complications.

The majority of studies concur that SPLH can be safely applied to treat not only benign, but also malignant liver disease [18,19,20,21,22,23]. Resection margins can be adequately achieved with this type of minimally invasive liver resection. This criteria was also met in the current study [19,21]. In recently published studies, authors analyze the long-term outcomes in HCC patients undergoing minimally invasive liver surgery. No significant difference in long-term survival and recurrent-free survival between SPLH and MPLH groups has been noted [21]. In the beginning of our experience with SPLH, this modality was mainly selected for patients with benign liver lesions without cirrhosis, as the single-port method is challenging by itself. As our experience grew, we began to include patients with colorectal metastases, and now we also include patients with hepatocellular carcinoma and underlying cirrhosis. Up to the year 2024, we have successfully treated two patients with hepatocellular carcinoma in cirrhotic liver; the operations were carried out with adequate resection margins and minimal blood loss.

Single-incision laparoscopic surgery has been introduced as a treatment modality for cholecystectomy, appendectomy, and moreover for colorectal surgery and laparoscopic surgery for gynecological diseases [30,31,32]. This advanced laparoscopic surgery technique mainly gained popularity among the surgical community due to presenting reduced tissue trauma and even better cosmetic results compared to multiport laparoscopic surgery [30,31,32]. Despite excellent immediate results, the biggest and most discussed issue remains the possible greater incidence of incisional hernia in single-incision laparoscopic surgery groups. One of the biggest challenges in reporting the true rate of incisional hernias in laparoscopic surgery is the fact that a long follow-up is required for an accurate rate to be reported [30,31,32]. In recent years, one of the largest studies comparing single-incision laparoscopic appendectomy and cholecystectomy to multiport laparoscopic surgery was performed by Barutcu and associates. A complete follow up of 286 patients after single-incision laparoscopic appendectomy and cholecystectomy was performed with a mean follow-up time of 58.4 months. The recorded incisional hernia rate was 2.4%, and the majority of patients (85.7%) underwent cholecystectomy [30]. When analyzing the risk factors, obesity and pre-existing umbilical hernia showed a significant association with developing incisional hernia. Contrary to the results in the presented cohort, where the incidence of incisional hernia was associated with the presence of biliary leak due to the presence of acute inflammation in the abdominal cavity, in the study of Barutcu, acute inflammation was not statistically significantly associated with the incidence of incisional hernia [29]. Moreover, in the study of Tschann and associates, where the incidence of incisional hernia was analyzed in single-incision laparoscopic colorectal surgery, the reported incisional hernia rate was 6%, with obesity and pre-existing umbilical hernia shown as significant risk factors for hernia development [31].

The biggest limitations of this study are its retrospective form and small sample size. As SPLH is a complex and technically difficult procedure, and therefore a longer learning curve is expected, appropriate selection of patients is crucial in the early phase of the learning curve. As a result of this, only a small number of patients were suitable for SPLH. With gained experience and comparable results to larger studies, we are confident that in the future, SPLH will be offered to more patients with both benign and malignant liver disease. As the literature and our experience suggest, the method can be safely applied for left lateral sectionectomy and partial hepatectomy for the fifth and sixth liver segments [19,21].

## 5. Conclusions

SPLH is a safe and feasible technique for the resection of benign and malignant liver lesions when performed by skilled and well-trained hepatobiliary surgeons. Although the presented cohort group is small, with specific patient selection, the authors believe that, compared to conventional open liver surgery and even MPLH, the operation times and length of hospitalization are shorter. Moreover, aesthetic outcomes are better, especially compared to open surgery. With the practice and further development of surgical instruments, this minimally invasive surgical technique has potential to become even more established in the field of laparoscopic and robotic liver surgery, especially for the treatment of lesions in the left lateral and anterior segments.

## Figures and Tables

**Figure 1 diseases-13-00187-f001:**
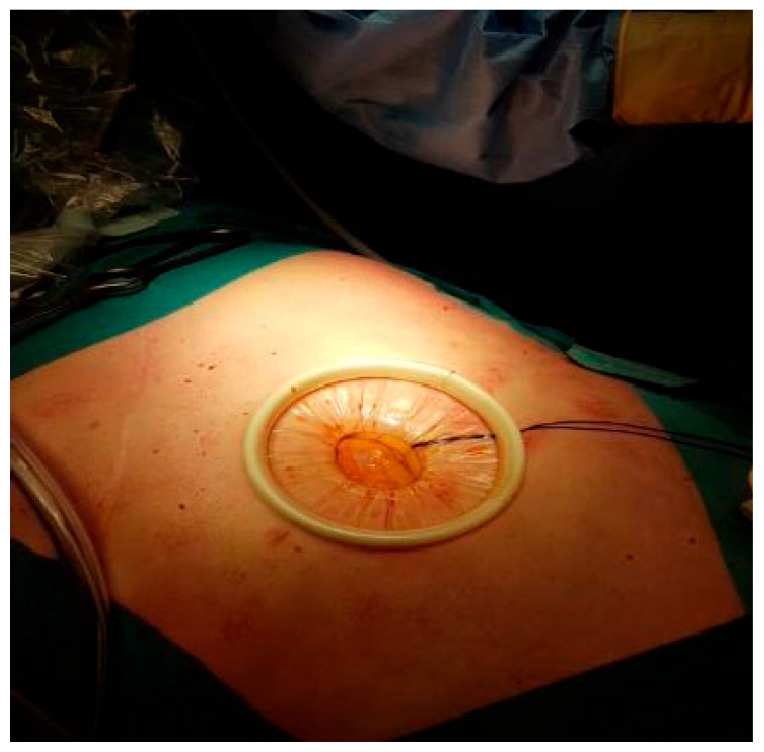
Supraumbilical midline laparotomy, prepared for the placement of the single-port platform.

**Figure 2 diseases-13-00187-f002:**
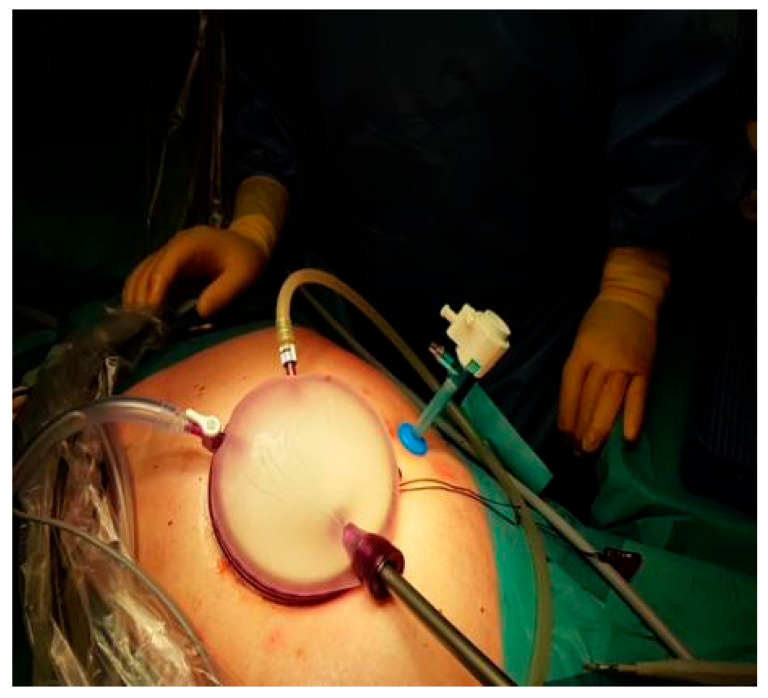
Placement of the GelPoint advanced access platform with the additional 5 mm port.

**Table 1 diseases-13-00187-t001:** Clinical features of the study population.

Clinical Features	
Diagnosis	
N (%)	
Colorectal metastases	3 (11.5)
Intrahepatic cholangiocarcinoma	1 (3.8)
Hepatocellular carcinoma	2 (7.7)
GIST	1 (3.8)
Caroli disease	1 (3.8)
Hydatid cyst	2 (7.7)
Simple biliary cyst	10 (38.5)
Hemangioma	3 (11.5)
Adenoma	3 (11.5)
Lesion location	
N (%)	
Segment 6	4 (15.4)
Segment 8	2 (7.7)
Segment 4b	3 (11.5)
Left lateral section	17 (65.4)
Previous abdominal operations	
N (%)	12 (46)
BMI [kg/m^2^]	
Mean ± SD (range)	26.6 ± 3.3 (20.8–34.6)

**Table 2 diseases-13-00187-t002:** Procedural details of the study population.

Procedure	
N (%)	
Subsegmentectomy	7 (27)
Segmentectomy	1 (3.8)
Bisegmentectomy	6 (23)
Deroofing ± cholecystectomy	10 (38.5)
Pericystectomy	2 (7.7)
Adhesions	
N (%)	7 (27)
Additional working port	
N (%)	
none	3 (11.5)
5 mm	19 (73.1)
12 mm	4 (15.4)

## Data Availability

All available data are presented in the case.

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
