# Peer review of "Single-Port Laparoscopic Hepatectomy: Slovenian Single-Center Experience"

_diseases, 2025, doi:10.3390/diseases13060187_

Round 1
Reviewer 1 Report
Comments and Suggestions for Authors
The manuscript titled “Single-Port Laparoscopic Hepatectomy: Slovenian Single Center Experience” presents a retrospective analysis of 26 patients undergoing single-port laparoscopic hepatectomy (SPLH) for benign and malignant liver lesions. The authors aim to evaluate the feasibility, safety, and outcomes of SPLH in a single high-volume tertiary center and compare their experience with existing literature.
The topic is timely and relevant, given the ongoing trend toward minimizing surgical invasiveness in hepatobiliary surgery. The manuscript demonstrates that SPLH, in experienced hands and carefully selected patients, is a technically feasible and safe alternative to multiport laparoscopic or open hepatectomy for lesions located in favorable segments (primarily segments II, III, and VI). The study reports promising perioperative outcomes, no conversions to open, and a low major complication rate (7%), limited to two cases of biliary leak. These findings are in line with previously published small-to-medium cohort studies.
The manuscript has several areas that would benefit from improvement before publication:
-
The manuscript contains numerous grammatical inaccuracies, awkward phrasings, and stylistic inconsistencies. Examples include repeated misuse of terms such as “therefor” instead of “therefore”, "troacar" and unclear sentence constructions. A thorough revision by a native English-speaking editor or a professional language editing service is strongly recommended to enhance readability and precision.
-
While the study is clearly based on real-world data and solid clinical experience, the lack of a control group (e.g., multiport or open hepatectomy) limits the generalizability. The author acknowledge this in the "limitations". The discussion could benefit from a comparison to the published cohorts (what's new?)
-
-
he tabular presentation of patient characteristics and outcomes is comprehensive but visually dense and difficult to interpret. The large table format spanning multiple pages would benefit from being split into separate, thematically focused tables (e.g., patient characteristics, procedural details, postoperative outcomes). Use of visual aids such as graphs or charts might further enhance data accessibility.
-
-
The surgical technique section is adequately described but would benefit from greater detail on intraoperative decision-making—particularly regarding the use of additional ports and management of adhesions. Inclusion of representative intraoperative images might appeal to the readership.
-
One point that merits further clarification is the choice of the GelPOINT advanced access platform as the standard access device for all procedures. While it is a widely used system in single-port surgery, the manuscript does not elaborate on why GelPOINT was favored over the single-incision platforms recommended by Intuitive.
-
The manuscript states that parenchymal transection was performed using LigaSure and CUSA, both of which are high-cost instruments. However, the rationale for choosing this combination is not discussed. In many centers, liver parenchyma is safely and effectively transected using standard bipolar and monopolar electrocautery, which are considerably more cost-efficient. Given that one of the main motivations for minimally invasive approaches is often improved resource utilization, a comment on the cost-effectiveness of the chosen instruments would be highly relevant.
-
The methods section mentions that “three trocars were inserted through the platform”, which raises a technical question: how were the necessary instruments managed during SPLH, considering that a standard single-port procedure typically requires a camera plus three working instruments (i.e. four), and often a additional channel is needed for optimal retraction or suction? A clarification of the port configuration—perhaps with a schematic drawing or intraoperative image—would be very helpful. If only three trocars were used, how was visualization and instrument handling optimized?
-
-
Author Response
Comment 1: The manuscript contains numerous grammatical inaccuracies, awkward phrasings, and stylistic inconsistencies. Examples include repeated misuse of terms such as “therefor” instead of “therefore”, "troacar" and unclear sentence constructions. A thorough revision by a native English-speaking editor or a professional language editing service is strongly recommended to enhance readability and precision.
Response: Thank you for your feedback, we revised the manuscript accordingly.
Comment 2: While the study is clearly based on real-world data and solid clinical experience, the lack of a control group (e.g., multiport or open hepatectomy) limits the generalizability. The author acknowledge this in the "limitations". The discussion could benefit from a comparison to the published cohorts (what's new?)
Response: Results of the SPLH were compared to the results of MPLH group as described in many paragraphs in Discussion section, for example paragraphs 194-222.
Comment 3: he tabular presentation of patient characteristics and outcomes is comprehensive but visually dense and difficult to interpret. The large table format spanning multiple pages would benefit from being split into separate, thematically focused tables (e.g., patient characteristics, procedural details, postoperative outcomes). Use of visual aids such as graphs or charts might further enhance data accessibility.
Response: Thank you for your valuable comment. The tables were rearranged accordingly (Table section in the manuscript).
Comment 5:The surgical technique section is adequately described but would benefit from greater detail on intraoperative decision-making—particularly regarding the use of additional ports and management of adhesions. Inclusion of representative intraoperative images might appeal to the readership.
Response: The addition of the figures and more detailed description of the operative technique was made (paragraphs 92-114).
Comment 6: One point that merits further clarification is the choice of the GelPOINT advanced access platform as the standard access device for all procedures. While it is a widely used system in single-port surgery, the manuscript does not elaborate on why GelPOINT was favored over the single-incision platforms recommended by Intuitive.
Response: thank you for the valuable point. Intuitive platform is not available at our institution. We added the clarification for the use of GelPoint in paragraphs 93-94.
Comment 7: The manuscript states that parenchymal transection was performed using LigaSure and CUSA, both of which are high-cost instruments. However, the rationale for choosing this combination is not discussed. In many centers, liver parenchyma is safely and effectively transected using standard bipolar and monopolar electrocautery, which are considerably more cost-efficient. Given that one of the main motivations for minimally invasive approaches is often improved resource utilization, a comment on the cost-effectiveness of the chosen instruments would be highly relevant
Response: The reasoning behind the use of CUSA and Ligasure was added in the paragraphs 105-110.
Comment 8: The methods section mentions that “three trocars were inserted through the platform”, which raises a technical question: how were the necessary instruments managed during SPLH, considering that a standard single-port procedure typically requires a camera plus three working instruments (i.e. four), and often a additional channel is needed for optimal retraction or suction? A clarification of the port configuration—perhaps with a schematic drawing or intraoperative image—would be very helpful. If only three trocars were used, how was visualization and instrument handling optimized?
Response: The description of the operative technique was expanded )paragraph 97-103). Additionally figures for easier clarification were added (under the Figure section).
Reviewer 2 Report
Comments and Suggestions for Authors
- Begin the introduction by providing a balanced overview of conventional laparoscopic surgery, addressing its advantages and disadvantages.
- Emphasize the clinical significance of Single-Port Laparoscopic Hepatectomy in decreasing the probability of post-operative complications, minimizing the hospital stay duration and Improving post-operative pain?
- In the method section, specify the qualifications and clinical expertise of the operating surgeon (e.g., hepatobiliary specialist witha postgraduate or doctorate), to enhance the reproducibility and credibility of the surgical approach.
- Briefly discuss the potential postoperative complications associated with hepatectomy in the introduction section.
- Identify and elaborate on patient-related and procedure-related factors known to affect the outcomes of Single-Port Laparoscopic Hepatectomy, such as tumor location, liver function status and surgeon experience.
- Include intraoperative median blood loss and discuss its relevance in the discussion section, particularly in relation to the minimally invasive nature of Single-Port Laparoscopic Hepatectomy.
- It is recommended to provide complete blood count and liver function tests results to access the physiological impacts of the procedure.
- Ensure consistency in the use of abbreviations throughout the manuscript.
- Provide a brief overview of current alternative treatments for benign and malignant liver tumors, including targeted therapy and immunotherapy. Discuss their relative efficacy and compare them with Single-Port Laparoscopic Hepatectomy.
- Do not repeat same information in multiple sections.
- Discussion section should include an analysis of postoperative recovery of patients after Single-Port Laparoscopic Hepatectomy including morbidity profiles, liver function, and quality of life.
- Address the technical limitations, like limited instrument manipulation and exposure of the surgical field.
- Conclusion should summarize the results and potential factors contributing these outcomes.
- Expand on future perspectives or clinical implications.
Author Response
Comment 1: Begin the introduction by providing a balanced overview of conventional laparoscopic surgery, addressing its advantages and disadvantages
Response: Thank you for your input. We addressed the topic in the Introduction segment, paragraphs 50-55 and 59-63.
Comment 2: Emphasize the clinical significance of Single-Port Laparoscopic Hepatectomy in decreasing the probability of post-operative complications, minimizing the hospital stay duration and Improving post-operative pain?
Response: The benifits of SPLH over the MPLH are discussed in paragraphs 189-201.
Comment 3: In the method section, specify the qualifications and clinical expertise of the operating surgeon (e.g., hepatobiliary specialist witha postgraduate or doctorate), to enhance the reproducibility and credibility of the surgical approach.
Response: thank you for your valuable input. The additional information regarding the surgeon skills were added in the paragraphs 85-86.
Comment 4: Briefly discuss the potential postoperative complications associated with hepatectomy in the introduction section.
Response: Changes were made accordingly in the paragraphs 59-63.
Comment 5: Identify and elaborate on patient-related and procedure-related factors known to affect the outcomes of Single-Port Laparoscopic Hepatectomy, such as tumor location, liver function status and surgeon experience.
Response: Changes were made accordingly in the Discussion section, paragraphs 176-181.
Comment 6: Include intraoperative median blood loss and discuss its relevance in the discussion section, particularly in relation to the minimally invasive nature of Single-Port Laparoscopic Hepatectomy.
Response: As the blood loss was already described in the Result section (paragraph 137), we added discussion on the topic in the paragraphs 198-204.
Comment 7: It is recommended to provide complete blood count and liver function tests results to access the physiological impacts of the procedure.
Response; Thank you for your input. As the intraoperative blood loss was small, the hemoglobin levels in our cohort were in the normal range throughout the hospitalisation. Therefore we believe that provision of complete blood count is unnecessary. Also, as only one patient had cirrhosis (Child A) and the resection of the liver was limited to two or less segments, no abrormal liver function test were noted among the study population. Therefore, this informations were not added to the text-
Comment 8: Ensure consistency in the use of abbreviations throughout the manuscript.
Response: Thank you for your input, the manuscript was corrected accordingly.
Comment 9: Provide a brief overview of current alternative treatments for benign and malignant liver tumors, including targeted therapy and immunotherapy. Discuss their relative efficacy and compare them with Single-Port Laparoscopic Hepatectomy.
Response: The authors believe that the discussion of the alternative treatmets is out of scope for this manuscript.
Comment 10: Do not repeat same information in multiple sections.
Response: The manuscript was corrected accordingly
Comment 11: Discussion section should include an analysis of postoperative recovery of patients after Single-Port Laparoscopic Hepatectomy including morbidity profiles, liver function, and quality of life.
Response: Thank you for your valuable input. The discussion of postoperative recovery was added in the paragraphs 198-206.
Comment 12: Address the technical limitations, like limited instrument manipulation and exposure of the surgical field.
Response: Thank you for the input, we added the discussion regarding the technical challenges in the paragraphs 175-193.
Comment 13: Conclusion should summarize the results and potential factors contributing these outcomes.
Response: The revision was made accordingly (paragraphs 281-282).
Comment 14: Expand on future perspectives or clinical implications.
Response: The expansion was made accordingly (paragraphs 281-282).
Round 2
Reviewer 1 Report
Comments and Suggestions for Authors
All issues raised by the reviewers have been adequately addressed.